# Clinical characteristics, systemic complications, and in-hospital outcomes for patients with COVID-19 in Latin America. LIVEN-Covid-19 study: A prospective, multicenter, multinational, cohort study

Luis F. Reyes[1,2,3]*, Alirio Bastidas[1], Paula O. Narváez[1], Daniela Parra-Tanoux[1], Yuli V. Fuentes[1,2], Cristian C. Serrano-Mayorga[1,2], Valentina Ortíz[1], Eder L. Caceres[1,2], Gustavo Ospina-Tascon[4,5], Ana M. Díaz[6], Manuel Jibaja[6], Magdalena Vera[7], Edwin Silva[1,8], Luis Antonio Gorordo-Delsol[9], Francesca Maraschin[3], Fabio Varón-Vega[10], Ricardo Buitrago[1,8], Marcela Poveda[1,8], Lina M. Saucedo[8], Elisa Estenssoro[11], Guillermo Ortíz[12], Nicolás Nin[13], Luis E. Calderón[4], Gina S. Montaño[1], Aldair J. Chaar[1], Fernanda García[6], Vanessa Ramírez[6], Fabricio Picoita[6], Cristian Peláez[6], Luis Unigarro[6], Gilberto Friedman[14], Laura Cucunubo[10], Alejandro Bruhn[7], Glenn Hernández[7], Ignacio Martin-Loeches[15,16], for the LIVEN-Covid-19 Investigators[¶]

1 Universidad de La Sabana, Chía, Colombiaa, 2 Department of Intensive Care, Clínica Universidad de La Sabana, Chía, Colombia, 3 Nuffield Department of Medicine, University of Oxford, Oxford, United Kingdom, 4 Department of Intensive Care, Fundación Valle del Lili, Cali, Colombia, 5 TransLab- CCM, Universidad Icesi, Cali, Colombia, 6 Critical Care Unit, Hospital Eugenio Espejo-Escuela de Medicina de la Universidad International, Quito, Ecuador, 7 Departamento de Medicina Intensiva, Facultad de Medicina, Pontificia Universidad Católica de Chile, Santiago, Chile, 8 Fundación Clínica Shaio, Bogota, Colombia, 9 Critical Care Unit, Hospital Juárez de México, Ciudad de México, México City, México, 10 Department of Intensive Care, Fundación Neumológica Colombiana-Fundación Cardioinfantil, Bogotá, Colombia, 11 Hospital Interzonal de Agudos San Martín de La Plata, La Plata, Argentina, 12 Universidad del Bosque, Bogotá, Colombia, 13 Intensive Care Unit, Hospital Español, Montevideo, Uruguay, 14 School of Medicine, Universidade de Federal do Rio Grande do Sul, Porto Alegre, Brazil, 15 Department of Clinical Medicine, St James's Hospital, Multidisciplinary Intensive Care Research Organization (MICRO), Dublin, Ireland, 16 Hospital Clinic, IDIBAPS, Universidad de Barcelona, CIBERes, Barcelona, Spain

¶ The complete list of LIVEN-Covid-19 Investigators is listed in the Acknowledgments.
* luis.reyes5@unisabana.edu.co

## Abstract

### Purpose

The COVID-19 pandemic has spread worldwide, and almost 396 million people have been infected around the globe. Latin American countries have been deeply affected, and there is a lack of data in this regard. This study aims to identify the clinical characteristics, in-hospital outcomes, and factors associated with ICU admission due to COVID-19. Furthermore, to describe the functional status of patients at hospital discharge after the acute episode of COVID-19.

### Material and methods

This was a prospective, multicenter, multinational observational cohort study of subjects admitted to 22 hospitals within Latin America. Data were collected prospectively. Descriptive

**Data Availability Statement:** All relevant data are within the paper and its Supporting information files.

**Funding:** The author(s) received no specific funding for this work.

**Competing interests:** The authors have declared that no competing interests exist.

statistics were used to characterize patients, and multivariate regression was carried out to identify factors associated with severe COVID-19.

## Results

A total of 3008 patients were included in the study. A total of 64.3% of patients had severe COVID-19 and were admitted to the ICU. Patients admitted to the ICU had a higher mean (SD) 4C score (10 [3] vs. 7 [3]), p<0.001). The risk factors independently associated with progression to ICU admission were age, shortness of breath, and obesity. In-hospital mortality was 24.1%, whereas the ICU mortality rate was 35.1%. Most patients had equal self-care ability at discharge 43.8%; however, ICU patients had worse self-care ability at hospital discharge (25.7% [497/1934] vs. 3.7% [40/1074], p<0.001).

## Conclusions

This study confirms that patients with SARS CoV-2 in the Latin American population had a lower mortality rate than previously reported. Systemic complications are frequent in patients admitted to the ICU due to COVID-19, as previously described in high-income countries.

## Introduction

Severe Acute Respiratory Syndrome Coronavirus 2 (SARS-CoV-2) is an RNA virus responsible for causing the new coronavirus disease 2019 (COVID-19), declared as a pandemic on March 11, 2020 [1]. This worldwide disease has shaken healthcare systems around the globe, causing more than 396 million infections and more than 5 million deaths [2]. It is estimated that the cost of in-hospital care of COVID-19 patients in the United States was between $9.6 billion and $16.9 billion in 2020. This approximation suggests an unprecedented burden on the countries' economies. It is known that a third of in-hospital care patients will develop severe Covid-19 and will require admission to the intensive care unit (ICU) [3–5]. The mortality rate of patients infected with SARS-CoV-2 that require hospital admission ranges between 3% and 88%, being higher in those admitted to the ICU [6, 7]. The main characteristics of patients who develop severe COVID-19 are older age, male, obesity, and several comorbid conditions [8–10]. Strikingly, these clinical characteristics and outcomes have been described in high-income countries [6, 11–14].

Latin America has been profoundly affected by the COVID-19 pandemic. In this region, socioeconomic contrasts are quite profound, with under-resourced healthcare systems and high poverty rates [5, 15]. Currently, just a few single countries in Latin America have described patients' clinical characteristics and clinical outcomes of patients admitted to the hospital due to COVID-19 [16]. Thus, there is scarce data describing patients with severe COVID-19, clinical features, and risk factors to develop severe illness in Latin America. Additionally, there is limited information concerning the functional status of these patients at hospital discharge. This study will attempt to provide novel data in this regard.

We hypothesize that COVID-19 has worse outcomes in the Latin American population. As a result, this study aims to describe the clinical features, systemic complications, factors associated with ICU admission due to COVID-19, and functional status at hospital discharge of patients with COVID-19 hospitalized in 8 Latin American countries.

## Materials and methods

This is an observational, prospective cohort study of subjects admitted to 22 hospitals due to SARS-CoV-2 infection in eight countries in Latin America between March 2020 and January 2021. These patients were included in a voluntary registry created by the Latin American Intensive Care Network (https://www.redliven.org). Data were collected prospectively by the attending physicians through reviewing medical records, laboratory data, and radiological images.

The main goal of this study was to identify the clinical characteristics, in-hospital outcomes, and factors associated with ICU admission due to COVID-19 in patients hospitalized in eight countries in Latin America. The secondary purpose of this study is to describe the frequency of systemic complications and the functional status of patients at hospital discharge after the acute episode of COVID-19.

### Participants

The cohort includes all patients hospitalized in general wards and ICU due to SARS-CoV-2 infection during the study period. All patients included in the study had confirmed SARS-CoV-2 infection determined by reverse transcription-polymerase chain reaction (rt-PCR) in a respiratory sample. All patients included in the cohort were analyzed in this study.

### Variable definitions

The complete definitions of the variables used in the study were provided to researchers in the study protocol before data collection. The 4C score was calculated using the data provided by each center. It includes the following variables on hospital admission: age, sex, number of comorbid conditions, respiratory rate, peripheral oxygen saturation, Glasgow coma scale, urea, and C-reactive protein [17, 18]. Acute respiratory distress syndrome was defined according to the Berlin classification using the Po2/Fio2 ratio, chest x-ray, and confirming non-cardiogenic etiology of the pulmonary affection [19]. Patients were stratified as obese when the body mass index was greater than 30. Advanced ventilatory support was defined as patients requiring invasive mechanical ventilation, non-invasive mechanical ventilation, or high flow nasal cannula. Physiological variables and laboratory results were gathered during the first 24 hours of hospital admission. According to the World Health Organization, self-care ability is defined as the ability of individuals to promote health, prevent disease, maintain health, and cope with illness and disability with or without the support of a healthcare provider [20–22]. The complete list of definitions is in the supplemental material.

### Data collection

Investigators from the eight countries collected prospective data using the case report form (CRF) built for this study on Research Electronic Data Capture (REDCap, version 8.11.11, Vanderbilt University, Nashville, Tenn.) [23] hosted by the Universidad de La Sabana, Chía, Colombia. The following variables were recorded in the CRF: age, gender, ethnicity, symptoms, comorbid conditions, physiological variables collected during the first 24 hours of hospital admission, chronic medications and treatments initiated in the ICU during the first 24 hours of hospital admission (e.g., requirement of advanced ventilatory support, vasopressor/inotropes usage), systemic complications, organ failure, and country of recruitment. Only patients with a reported hospital discharge date were included in calculating the hospital length of stay and mortality rates.

## Statistical analysis

Discrete variables are expressed as frequencies and percentages. Continuous variables with normal distribution are expressed as means (standard deviation); variables with no normal distribution are expressed as median (interquartile ranges). Categorical variables are presented in counts (percentages) and were evaluated through the Chi-square test. For continuous variables with normal distribution, the t Student test was performed, and for variables with no normal distribution Wilcoxon-Mann-Whitney test was used. Multivariate logistic regression was performed to determine those factors associated with ICU admission due to COVID-19. To apply this model, variables that had supporting literature, biological plausibility, and a p-value of <0.2 in the bivariate analysis were included in the model. Statistical significance was set at $p<0.05$. All statistical analysis was carried out in IBM SPSS 27 for MAC.

## Results

A total of 3008 patients with confirmed SARS-CoV-2 infection were included in the study. Most patients were enrolled from lower-middle-income countries (90.2%, 2715/3008). Most patients were enrolled in Colombia (67%, 2027/3008), followed by Ecuador (15.4%, 465/3008) and Chile (9.7%, 293/3008) (Fig 1). A total of 64.3% (1934/3008) were admitted to the ICU (Table 1).

## Demographic and clinical characteristics

Patients were mainly male (60.4%, 1817/3008), with a median (IQR) age of 56 (43–67) years old. Many patients included in the study had comorbid conditions, arterial hypertension being the most frequently identified (34.6%, 1041/3008), followed by obesity (24.9%, 748/3008), non-complicated diabetes mellitus (14.8%, 445/3008), and chronic pulmonary disease (7.7% 232/3008), among others (Table 1). Several differences were observed between patients admitted to the ICU and those treated in general wards. For instance, the cumulative frequency of ICU admission increased in direct proportion to age (median [IQR]) (47.9 [17.5] vs 58.7 [14.6] $p = 0.001$) (Fig 1). Other common comorbidities seem more frequently in ICU patients were chronic arterial hypertension (42.7% [825/1934] vs. 20.1% [210/1074], $p<0.001$), obesity (33.2% [642/1934] vs. 9.9% [106/1074], $p<0.001$), chronic pulmonary disease (9.0% [174/1934] vs. 5.4% [58/1074], $p<0.001$), and chronic kidney disease (8.4% [163/1934] vs. 2.6% [28/1074], $p<0.001$), among others (Table 1).

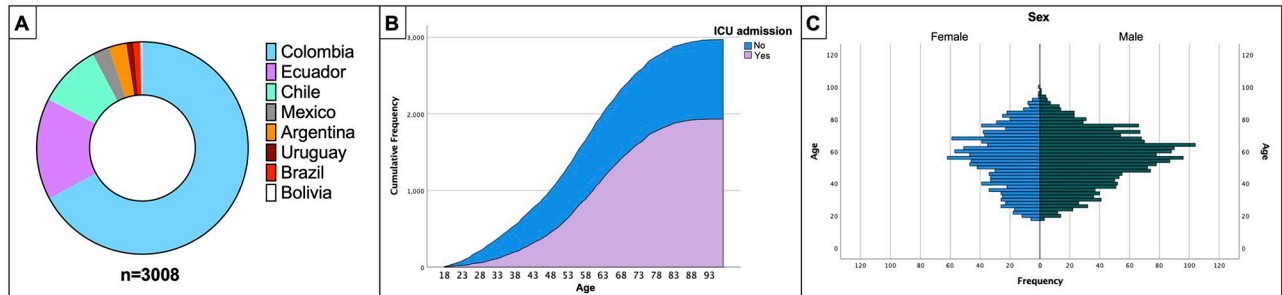

**Fig 1. Older patients get sicker and have a higher cumulative frequency of ICU admission.** (A) The proportion of patients enrolled in the study per country. (B) The figure presents the cumulative number of cases included in the study; in purple, patients admitted to the Intensive Care Unit (ICU) and patients with no admission to the ICU in blue. (C) The age distribution of subjects in the study is shown in this figure. Age ranges are listed down the center of the graph, and sex distribution is displayed on each side.

**Table 1. Baseline characteristics of patients with confirmed SARS-CoV-2 infection that developed severe COVID-19 stratified by patients admitted to the Intensive Care Unit (ICU).**

| Characteristic | All | Patients admitted to the ICU | | p-value |
| | | No | Yes | |
| | n = 3008 | n = 1074 | n = 1934 | |
| --- | --- | --- | --- | --- |
| **Demographics** | | | | |
| Age, median (IQR) | 56.0 (43–75) | 47.9 (17.5) | 58.7 (14.6) | **<0.001** |
| Female, **n (%)** | 1191 (39.6%) | 547 (50.9%) | 644 (33.3%) | **<0.001** |
| **Chronic comorbid conditions, n (%)** | | | | |
| Cardiovascular Disease | 277 (9.2) | 62 (5.8) | 215 (11.1) | **<0.001** |
| Chronic Arterial Hypertension | 1041 (34.6) | 216 (20.1) | 825 (42.7) | **<0.001** |
| Chronic Pulmonary Disease | 232 (7.7) | 58 (5.4) | 174 (9.0) | **<0.001** |
| Asthma | 47 (1.6) | 26 (2.4) | 21 (1.1) | 0.005 |
| Non-Complicated Diabetes Mellitus | 445 (14.8) | 57 (5.3) | 388 (20.1) | **<0.001** |
| Complicated Diabetes Mellitus | 182 (6.2) | 21 (2.0) | 164 (8.5) | **<0.001** |
| Obesity | 748 (24.9) | 106 (9.9) | 642 (33.2) | **<0.001** |
| Chronic Neurological Disorder | 73 (2.4) | 25 (2.3) | 48 (2.5) | 0.792 |
| Chronic Kidney Disease | 191 (6.3) | 28 (2.6) | 163 (8.4) | **<0.001** |
| Malignant neoplasm | 80 (2.7) | 22 (2.0) | 58 (3.0) | 0.121 |
| AIDS/HIV | 15 (0.5) | 8 (0.7) | 7 (0.4) | 0.153 |
| **Past medical history, n (%)** | | | | |
| Pregnancy | 17 (1.2) | 13 (2.5) | 4 (0.5) | **<0.001** |
| Smoking | 235 (7.8) | 52 (4.8) | 183 (9.5) | **<0.001** |
| Healthcare worker | 133 (4.4) | 102 (9.5) | 31 (1.6) | **<0.001** |
| **Symptoms on admission, n (%)** | | | | |
| Fever | 1861 (61.9) | 546 (50.8) | 1315 (68.0) | **<0.001** |
| Cough—productive | 618 (20.5) | 115 (10.7) | 503 (26.0) | **<0.001** |
| Rhinorrhea | 260 (8.6) | 131 (12.2) | 129 (6,7) | **<0.001** |
| Wheezing | 80 (2.7) | 9 (0.8) | 71 (3.7) | **<0.001** |
| Chest Pain | 381 (12.7) | 142 (13.2) | 239 (12.4) | 0.495 |
| Myalgia | 938 (31.2) | 290 (27.0) | 648 (33.5) | **<0.001** |
| Joint pain-arthralgia | 662 (22.0) | 153 (14.2) | 509 (26.3) | **<0.001** |
| Shortness of breath | 1776 (59.0) | 342 (31.8) | 1434 (74.1) | **<0.001** |
| Chest wall drawing | 100 (3.3) | 8 (0.7) | 92 (4.8) | **<0.001** |
| Headache | 876 (29.1) | 395 (36.8) | 481 (24.9) | **<0.001** |
| **Physiological parameters on admission, mean (SD)** | | | | |
| Systolic blood pressure, mmHg | 123.8 (29.9) | 123.4 (16.7) | 124 (23.1) | 0.441 |
| Diastolic blood pressure, mmHg | 72.0 (13.7) | 74.77 (1.7) | 70.38 (14.6) | **<0.001** |
| Glasgow | 11.9 (5.0) | 14 (0.8) | 9 (5.6) | **<0.001** |
| **Laboratories on hospital admission, mean (SD)** | | | | |
| *Arterial gases* | n = 2214 | n = 474 | n = 1738 | |
| Fraction of inspired oxygen (FiO2), % | 48.3 (29.7) | 22 (6.4) | 63 (26.9.) | **<0.001** |
| Bicarbonate (HCO3), mmol/L | 21.4 (4.2) | 20.99 (3.53) | 21.53 (4.49) | 0.016 |
| Lactate, mmol/L | 1.7 (1.3) | 1.44 (1.1) | 1.84 (1.4) | **<0.001** |
| *Complete Blood Count* | n = 2258 | n = 507 | n = 1751 | |
| Leucocytes, x10$^3$ cells | 10.5 (5.3) | 8.0 (3.5) | 11.2 (5.5) | **<0.001** |
| Lymphocytes, % | 12.4 (11.4) | 18.5 (12.8) | 10.5 (10.2) | **<0.001** |
| Neutrophiles, % | 69.4 (24.9) | 71.6 (16.4) | 68.6 (27.1) | 0.018 |
| Hematocrit, % | 41.2 (6.8) | 42.8 (6.0) | 40.7 (7.0) | **<0.001** |

(*Continued*)

**Table 1.** (Continued)

| Characteristic | Patients admitted to the ICU | | | p-value |
|---|---|---|---|---|
| | **All** | **No** | **Yes** | |
| | **n = 3008** | **n = 1074** | **n = 1934** | |
| Hemoglobin, g/dL | 13.7 (2.4) | 14.5 (2.1) | 13.5 (2.4) | **<0.001** |
| Platelets, x10³ cells | 251.4 (106.2) | 233.0 (95.5) | 256.7 (108.5) | **<0.001** |
| *Liver function tests* | n = 1658 | n = 310 | n = 1348 | |
| Bilirubin, mg/dL | 0.8 (1.0) | 0.7 (0.7) | 0.84 (1) | 0.115 |
| Alanine Transaminase (ALT), U/L | 52.3 (37.0) | 45 (32.2) | 54.1 (37.8) | **<0.001** |
| Aspartate Transaminase (AST), U/L | 56.5 (37.1) | 48.1 (31.1) | 58.5 (38.1) | **<0.001** |
| *Renal function tests* | n = 2192 | n = 455 | n = 1737 | |
| Ureic nitrogen, mg/dL | 26.8 (20.3) | 18.9 (12.6) | 28.9 (21.5) | **<0.001** |
| Serum creatinine, mg/dL | 1.35 (1.8) | 1 (1.41) | 1.4 (1.9) | **<0.001** |
| *Metabolic tests* | n = 1924 | n = 312 | n = 1612 | |
| Sodium (Na), mEq/L | 136.9 (5.0) | 136.7 (4.6) | 137 (5.1) | 0.427 |
| Potassium (K), mEq/L | 4.3 (0.7) | 4.26 (0.65) | 4.3 (0.7) | 0.362 |
| *Coagulation times* | n = 1347 | n = 85 | n = 1262 | |
| Prothrombin Time (PT), s | 14.9 (8.5) | 14.1 (6.5) | 15.0 (8.6) | 0.348 |
| Partial Thromboplastin Time (PTT), s | 33.1 (10.9) | 32.1 (9.6) | 33.2 (11.0) | 0.376 |
| International Normalized Ratio (INR) | 1.1 (0.3) | 1.1 (0.51) | 1.1 (0.3) | 0.463 |
| *Acute—Phase Reactants* | n = 1672 | n = 1293 | n = 379 | |
| C-reactive protein, mg/L | 68.3 (71.8) | 68.3 (62.2) | 68.3 (74.4) | 0.991 |
| **Disease severity** | | | | |
| 4C Score, mean (SD) | 9.2 (3.7) | 7 (3) | 10 (3) | **<0.001** |

**IQR**, interquartile range; **AIDS**, acquired immunodeficiency syndrome; **HIV**, Human Immunodeficiency Virus; IQR, Interquartile Range; SD, Standard Deviation.

The most commonly reported clinical symptoms on hospital admission were cough (76,9%, 2314/3008), fever (61,9%, 1861/3008), shortness of breath (59%, 1776/3008), and myalgia (31.2%, 938/3008) (Table 1). When comparing the symptoms of patients admitted to the ICU vs non-ICU patients, we found ICU patients more frequently presented with shortness of breath (74% [1434/1934] vs. 31.8% [342/1074], $p<0.001$), fever (68% [1315/1934] vs. 50.8% [546/1074], $p<0.001$), myalgia (33.5% [648/1934] vs. 27.0% [290/1074], $p<0.001$), arthralgias (26.3% [509/1934] vs. 14.2% [153/1074], $p<0.001$), and productive cough (26% [503/1934] vs. 10.7% [115/1074], $p<0.001$) (Table 1).

## Disease severity, in-hospital treatments, and systemic complications

When assessing disease severity, the 4C score was used. In patients admitted to the ICU, the Mean (SD) 4C score was higher than in non-ICU patients (10[3] vs. 7[3] $p<0.001$) (Table 1). A direct correlation between a higher 4C score and ICU admission rate was observed in Fig 2. The most commonly administered treatments in all cohorts were corticosteroids (54.5%, 1578/3008), systemic antibiotics (48.4%, 1456/3008), and vasopressors or inotropic agents (36.9%, 1111/3008). Invasive mechanical ventilation rate was higher in ICU admitted patients (71% [1391/1934] vs. 1.5% [16/1074], $p<0.001$) as expected. Tracheostomy was performed in 21% (293/1407) of the patients treated with invasive mechanical ventilation. ICU patients were recurrently treated with corticosteroids (69% [1334/1934] vs. 22.7% [244/1074], $p<0.001$) antibiotics (61.2% [1183/1934] vs. 25.4% [273/1074], $p<0.001$), vasopressors or inotrope agents

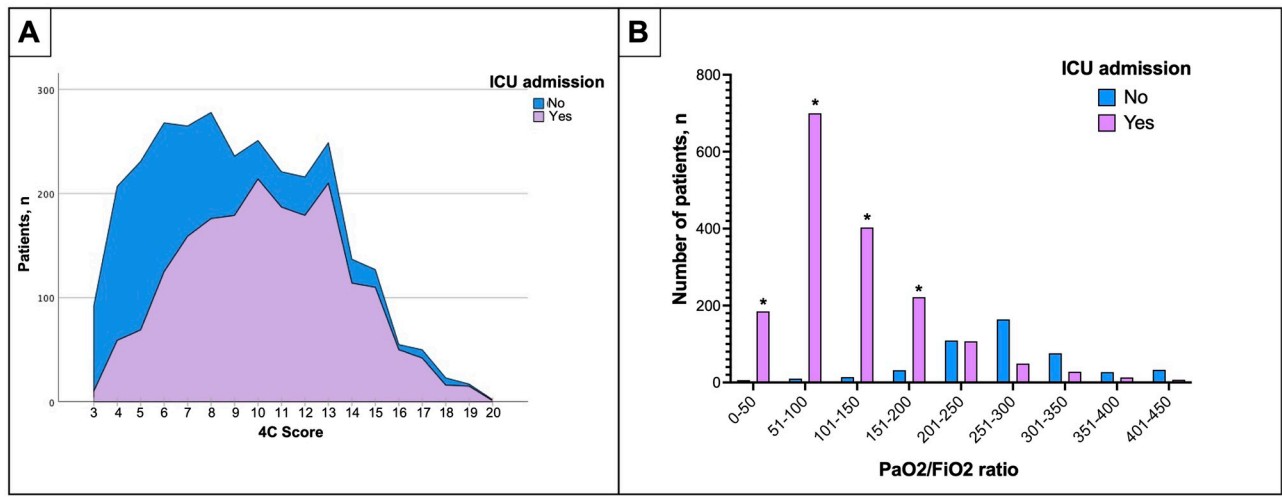

**Fig 2.** (A) Correlation between the 4C score and the ICU admission rate. This figure compares the number of patients admitted to the ICU (Y-axis) and their punctuation in the 4C score (X-axis). In purple, the patients were admitted to the ICU, and in blue, patients were not admitted. The ICU admission rate increases as the 4C score do. In contrast, most patients rated with 4C scores of 6 or less were more frequently treated outside the ICU. (B) Comparison between PaO2/FiO2 ratio and the number of patients admitted to ICU. This figure compares the number of patients admitted to ICU in purple columns the number of patients who were not admitted to the ICU in blue columns. Patients with a low PaO2/FiO2 ratio were the most admitted to ICU.

(57% [1100/1934] vs. 1% [11/1074], $p<0.001$) and dialysis (17.4% [337/1934] vs. 0.8% [9/1074], $p<0.001$) more than non-ICU patients (S1 Table).

Pulmonary complications were the most identified complications in our cohort. A total of 39.1% (1179/3008) of patients developed acute respiratory distress syndrome (ARDS). This was significantly higher in patients admitted to the ICU (56.2% [1087/1934] vs. 8.6% [92/1074], $p<0.001$). Additionally, 23.0% (692/3008) of patients developed acute kidney injury, 15.8% (476/3008) anemia, and 7.5% (225/3008) a cardiac arrhythmia; all these complications were repeatedly found more frequently in patients admitted to the ICU (S2 Table).

## Clinical outcomes

The in-hospital mortality reported in our cohort was 24.1% (725/3008). The in-hospital mortality rate in patients admitted to the ICU was 35,1% (678/1934) and 4,4% [(47/1074) $p<0.001$] in non-ICU patients with COVID-19. Regarding hospital length of stay (LOS), we only include 2823 patients because a total of 185 patients has missing data of discharge date; the overall median (IQR) observed in the cohort was 10 (4–19); when stratified by ICU admission, we found that ICU admitted patients had significantly longer hospital LOS (15 [9–26] vs. 3 [0–7], $p<0.001$). Finally, self-care at hospital discharge was evaluated. The majority of patients had equal self-care ability at discharge (43.8%, 1319/3008); however, patients admitted to the ICU had worse self-care ability at discharge when compared with non-ICU patients (25.7% [497/1934] vs. 3.7% [40/1074], $p<0.001$) (Table 2).

## Risks factors associated with ICU admission on COVID–19 patients: A multivariate analysis

After performing the multivariate analysis, we found an OR [95%CI], age (1.02 [1.00–1.03] $p = 0.019$), shortness of breath (3.04 [2.02–4.58] $p<0.001$), obesity (2.43 [1.45–4.07] $p = 0.001$),

**Table 2. Clinical outcomes.**

| Outcomes | All | Patients admitted to the ICU | | p-value |
| | | No | Yes | |
| | n = 3008 | n = 1074 | n = 1934 | |
|---|---|---|---|---|
| Mortality, n (%) | 725 (24.1) | 47 (4.4) | 678 (35.1) | **<0.001** |
| Referred to another hospital, n (%) | 243 (8.1) | 61 (5.7) | 182 (9.4) | 0.109 |
| Referred to palliative care Program, n (%) | 4 (0.1) | 2 (0.2) | 2 (0.1) | 0.550 |
| Ambulatory dialysis, n (%) | 20 (0.7) | 5 (0.5) | 15 (0.8) | 0.316 |
| **Length of stay** | **All** | **No** | **Yes** | |
| | n = 2823 | n = 1028 | n = 1795 | |
| Hospital LOS, median (IQR) | 10 (4–19) | 3 (0–7) | 15 (9–26) | **<0.001** |
| **Self-care at discharge** | | | | |
| Worst Self-care ability at discharge, n (%) | 537 (17.9) | 40 (3.7) | 497 (25.7) | **<0.001** |
| Equal Self-care ability at discharge, n (%) | 1319 (43.8) | 910 (84.7) | 409 (21.2) | **<0.001** |
| Better Self-care ability at discharge, n (%) | 70 (2.3) | 36 (1.9) | 34 (3.2) | 0.023 |

**LOS**, length of stay in days

increased serum lactate (1.74 [1.24–2.47] $p$ = 0.002), and leukocytosis (1.10 [1.05–1.17] $p<0.001$) were independently associated with ICU admission (Table 3).

## Discussion

This study describes the clinical characteristics, systemic complications, and outcomes from a prospective, multinational cohort of hospitalized patients diagnosed with COVID-19 from eight countries in Latin America. In our cohort, we found that age, shortness of breath, obesity,

**Table 3. Logistic binary multivariate analysis fitted to assess the factors associated with admission to the intensive unit (ICU).**

| Variable | OR | 95% CI | | p-value |
| | | Lower | Upper | |
|---|---|---|---|---|
| Age | 1.02 | 1.00 | 1.03 | **0.019** |
| Sex | 1.10 | 0.72 | 1.68 | 0.659 |
| Healthcare worker | 0.51 | 0.14 | 1.96 | 0.330 |
| Number of comorbid conditions | 0.99 | 0.83 | 1.21 | 0.983 |
| Shortness of breath | 3.04 | 2.02 | 4.58 | **<0.001** |
| Glasgow | 0.71 | 0.63 | 0.8 | **<0.001** |
| Obesity | 2.43 | 1.45 | 4.07 | **0.001** |
| Smoking | 1.47 | 0.72 | 3.03 | 0.290 |
| Diastolic blood pressure, mmHg | 0.98 | 0.97 | 1.00 | 0.015 |
| SaO2, % | 1.02 | 1.00 | 1.05 | 0.103 |
| Lactate, mmol/L | 1.74 | 1.24 | 2.47 | **0.002** |
| Leucocytes, x10$^3$ cells | 1.10 | 1.05 | 1.17 | **<0.001** |
| Lymphocytes, % | 0.96 | 0.95 | 0.98 | **<0.001** |
| Hematocrit, % | 0.95 | 0.93 | 0.99 | **0.003** |
| Platelets, x10$^3$ cells | 1.00 | 1.00 | 1.00 | 0.765 |
| Ureic nitrogen, mg/dL | 1.01 | 0.99 | 1.02 | 0.272 |
| Alanine transaminase, U/L | 1.01 | 1.00 | 1.02 | 0.149 |
| Aspartate transaminase, U/L | 1.00 | 0.99 | 1.01 | 0.593 |
| C-reactive protein, mg/L | 0.99 | 1.00 | 1.00 | 0.055 |

leukocytosis, and increased serum lactate were independently associated with ICU admission due to COVID-19. The most common complications in this cohort were ARDS, shock, and acute kidney injury. We identified that mortality rates and length of hospital stay were significantly higher in patients admitted to the ICU than those hospitalized in the general wards. Patients admitted to the ICU due to COVID-19 were found to have lower self-care capacity at hospital discharge, which might indicate long-term COVID-19 consequences.

Lower respiratory tract infections range from mild to severe, with varying degrees of systemic complications and COVID-19 is not the exception [8, 24, 25]. There are robust data linking older age, male sex, and obesity with a greater risk of developing severe COVID-19 [13, 26–28]. These risk factors have also been associated with mutations in the innate immune system in males, limiting the host capacity to generate a robust immune response when encountering the SARS-CoV-2 virus [29–31]. Our study also found that shortness of breath and elevated serum concentrations of lactate were independently associated with ICU admission. This is concordant with what is reported in the literature, as only severe COVID-19 patients are admitted to ICU [32, 33]. These factors are essential because they are easily identifiable by physicians on hospital admission and might guide them to early detection of patients at risk of developing severe disease.

COVID-19 patients develop systemic complications in up to 68% of cases. A metanalysis of 44 peer-reviewed studies, most of them from China and other Asian countries, including 14866 patients, found a prevalence of ARDS of 14%, acute cardiac injury (15%), and venous thromboembolism (15%) as the most frequent complications [34]. Another metanalysis that included 2874 patients described an ARDS frequency of 32.8% [35]. In our cohort, ARDS was the most common complication observed in 56.2% of subjects. Furthermore, cardiac injury in COVID-19 has also been described as showing higher mortality rates when present [36, 37]. Despite not being one of the most regularly seen complications in our study, we found a similar prevalence. In COVID-19 patients, the most described thrombotic complications are pulmonary embolism and deep venous thrombosis, as Shah *et al.* observed in a multicenter retrospective observational study. Thrombotic complications were documented in 47.7% of cases, 22.5% with pulmonary embolism, and 11.8% with deep vein thrombosis; the rest were arterial complications, including myocardial infarction [38]. Our study documented a very low frequency of documented pulmonary embolism, less than 1%. We believe that this low prevalence of pulmonary embolism could be associated with our study's real-world data, meaning that patients were not systematically screened for pulmonary embolism unless high clinical and laboratory suspicion. Only patients with radiological confirmation and clinical symptoms consistent with pulmonary embolism were reported.

In this multicenter study, acute kidney injury (AKI) has also been regularly documented. As previously described in several studies, it is a clear marker of worse clinical outcomes in COVID-19 patients. Silver *et al.* showed AKI prevalence is up to 46% of ICU patients [39]. Potere *et al.* reported a much lower prevalence (6%) in their metanalysis from mostly Asian studies [34]. Here, AKI was observed in 23% of patients in general wards and 34.2% of patients in the ICU. This difference in proportions could be attributed to the higher percentage of patients hospitalized in ICU than those in a regular ward.

Several studies have evaluated hospital and ICU mortality as a primary outcome in COVID-19 patients [40, 41]. The COVID-19 Lombardy ICU network reported an ICU mortality of 48.7% in a retrospective observational cohort including 3988 patients [27]. Petrilli *et al.* reported overall mortality in critically ill patients of 57% in a prospective cohort including 5279 patients from New York City [42]. Moreover, a meta-analysis including 37 articles revealed that the pool prevalence of ICU mortality in patients with COVID-19 was 32%. This meta-analysis did a subgroup analysis by the country where the highest mortality rates were

reported in China (42%), followed by the USA (36%) [43]. Our cohort found an ICU mortality rate of 35%, similar to the overall mortality presented on the metanalysis [43], though relatively lower when compared to the United States and the Italian cohorts. This is important because even though Latin American countries did not have robust ICU capacity before the COVID-19 pandemic, countries had almost three months to prepare after the pandemic began in China. Thus, we hypothesize that this lack of time might play a crucial role in this lower reported mortality.

Recently, there has been growing concern about the potential long-term complications in COVID-19 patients that survive acute infection [22, 44, 45]. The COMEBACK study group studied long-term complications using telephone interviews in a cohort of 478 COVID survivors in France. They found that approximately half of the patients remained with at least one symptom that was not present before the COVID-19 infection [46]. Moreover, Garrigues *et al.* conducted a single-center study including 120 patients hospitalized due to COVID-19. After a mean of 110.9 days following admission, found that persistent symptoms and lower health-related quality of life [47]. We found that self-care ability at discharge in our cohort significantly decreased in ICU admitted patients. However, it is unknown whether this lower functional capacity was exclusively associated with COVID-19 or post-ICU syndrome. However, these findings should alert healthcare providers to the potential necessity of creating follow-up clinics for COVID-19 survivors. The importance of long-term monitoring of patients after infection by SARS-CoV2 lies in the impact of persistent symptoms, worse quality of life, ability to work, and the possible need for rehabilitation programs. This should alert countries to the potential burden that COVID-19 could impose on healthcare systems and economies after the pandemic.

Our study has several strengths and limitations that are important to explore. Although we enrolled patients in more than twenty hospitals in eight Latin American countries, there were several Latin American countries that we did not include. Therefore, these results might not be generalizable to all of Latin America. Secondly, despite comparing ICU patients to those in general wards, our cohort was composed of patients with severe COVID-19. Thus, further studies including a more considerable proportion of patients with non-severe COVID-19 are essential. Finally, this study was not designed to assess the potential implications of long-COVID-19; thus, we cannot provide robust conclusions regarding this critical problem. However, we found that patients did have lower functional status at hospital discharge, which might serve as a hypothesis for future studies.

## Conclusions

Patients with COVID-19 admitted to ICU included in our Latin American multicenter study had a lower mortality rate than previously reported in the literature. Age, obesity, elevated serum lactate, leukocytosis, and shortness of breath on admission were independently associated with ICU admission in patients with COVID-19. Systemic complications are frequent in patients admitted to the ICU due to COVID-19, as previously described in high-income countries. Finally, patients admitted to ICU due to COVID-19 infection had lower self-care capacity than those not admitted to ICU. This might have significant long-term implications for Latin American countries.

## Supporting information

**S1 Table. Treatments stratified by patients admitted to the Intensive Care Unit (ICU).** (DOCX)

**S2 Table. Patients with severe COVID-19 that developed complications stratified by patients admitted to the Intensive Care Unit (ICU).**
(CSV)

**S1 File. Data set with the information recollected for this study.**
(PDF)

## Acknowledgments

**^LIVEN-Covid-19 Investigators** (To be included in PubMed as collaborators):

Elsa D. Ibañez-Prada, Laura Bravo, Paula Ramirez, Ingrid G. Bustos, Julian Lozada, Manuela Saenz-Valcarcel, Enrique Gamboa, Salome Gomez, Esteban Garcia-Gallo, Alfonso José Arango, Álvaro Aguilar, Andrea Lizeth Ayala, Andrea Viviana Bayona, Angelica Rodríguez, Carol Viviana Aponte, Carolina Forero-Carreño, Conny Stefanny Muñoz, Cristian Augusto Estrada, Cristopher Romero, Cristian Peláez, Danilo Trujillo, Diego Holguin, Fabricio Picoita, Jesús Chávez-Villegas, Faure Rodríguez, Francisco Franco, Fernanda García, Hernan Sánchez, Janett Vanessa Moncayo, Jennifer A. Pinedo, Jesica Valeria Bravo, José David Cruz, José Miguel Angel, Jovany Castro-Lara, Karen Andrea Mantilla, Lorena García, Lorena Pabón, Luis Arturo López-Reveles, Luis Fernando Mamani, Maria Gabriela Saenz, Cecilia Loudet, Nahuel Rubatto Birri, Luis Unigarro, Marisa Lucrecia Yupa, Valeria Catalina Quevedo, Vanessa Ramírez, Paola Sánchez, Hernán Sánchez, Jorge Antonio Caamaño Solis, Edgar Segundo Espinosa Morales, Edgar Segundo Espinosa Morales, Marco Antonio Jiménez Espinosa, Luis Eduardo Males Vinueza, Luis Fernando Martinez Arias, Juan Pablo Monge Casanova, Jose Andres Moreno Troya, Diego Rolando Morocho Tutillo, Hector Homero Moya, Ximena Alexandra Noboa Gallegos, Nelson Gustavo Remache Vargas, Milton Alfredo Tobar Galindo, Freddy Rogelio Sanchez, Liliana Elizabeth Torres Martinez, Amparo Rocío Basantes Sánchez, Monica Viviana Medina Cabrera, Diego Andres Mora Palma, Verónica Elizabeth Paredes Carvajal, Hernan Andres Sanchez Freire, Sayra de los Angeles Caiza Mosquera.

## Declarations

**Ethics approval and consent to participate**: This research is categorized as risk-free. It is a study where clinical data were obtained from an anonymized database. Thus, the informed consent was waived. The Institutional Review Board approved this study at Clinica Universidad de La Sabana and Universidad de La Sabana (MED-311-2021).

**Consent for publication**: All authors reviewed this manuscript and consented to its publication.

## Author Contributions

**Conceptualization:** Luis F. Reyes, Magdalena Vera, Luis Antonio Gorordo-Delsol.

**Data curation:** Luis F. Reyes, Edwin Silva.

**Investigation:** Yuli V. Fuentes, Elisa Estenssoro.

**Methodology:** Luis F. Reyes, Edwin Silva.

**Project administration:** Ana M. Díaz.

**Resources:** Elisa Estenssoro.

**Supervision:** Alirio Bastidas, Eder L. Caceres, Ricardo Buitrago, Lina M. Saucedo, Gina S. Montaño, Aldair J. Chaar, Cristian Peláez, Luis Unigarro, Laura Cucunubo, Glenn Hernández.

**Validation:** Luis F. Reyes, Gustavo Ospina-Tascon, Manuel Jibaja, Magdalena Vera, Fabio Varón-Vega, Marcela Poveda, Guillermo Ortíz, Nicolás Nin, Fernanda García, Vanessa Ramírez, Fabricio Picoita, Gilberto Friedman, Alejandro Bruhn, Ignacio Martin-Loeches.

**Visualization:** Ana M. Díaz, Luis E. Calderón.

**Writing – original draft:** Alirio Bastidas, Yuli V. Fuentes, Cristian C. Serrano-Mayorga, Valentina Ortíz.

**Writing – review & editing:** Luis F. Reyes, Paula O. Narváez, Daniela Parra-Tanoux, Yuli V. Fuentes, Cristian C. Serrano-Mayorga, Francesca Maraschin.

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
