## [Decision Letter · Decision Letter 0]

2 Feb 2022

PONE-D-21-34973

Clinical Characteristics, Systemic Complications, and In-Hospital Outcomes for Patients with COVID-19 in Latin America. LIVEN-Covid-19 Study: A Prospective, Multicenter, Multinational, Cohort Study.

PLOS ONE

We look forward to receiving your revised manuscript.

Kind regards,

Ezio Lanza, M.D.

Academic Editor

PLOS ONE

Journal Requirements:

Reviewers' comments:

Reviewer's Responses to Questions

Review Comments to the Author

Reviewer #1: In this prospective study, authors investigated the risk factors associated with ICU admission in COVID-19 patients and the outcomes (systemic complications, mortality, functional status at discharge) in hospitalized patients in 8 Latin American countries. Overall, the manuscript is sound and clearly written, and data are clearly presented in tables; however, authors could improve the manuscript according to the following suggestions.

Introduction:

- Add recent references in the first paragraph to update data about the mortality rate and main characteristics associated to severe COVID-19

Methods:

- Lines 120-122: add reference

- Lines 132-133: why did you consider only treatment initiated in the ICU during the first 24 hours of hospital admission in your analysis?

Results:

- Lines 200-202: specify how many patients you included in the LOS evaluation, according to the sentence in the Methods section (lines 134-135). Data should also be reported in a table.

Discussion

- Line 231: add references

- Line 268: add references

Table

- Table 1: Add CRP in the section of laboratories test on hospital admission

- Table 1: remove LOS from the acronym list because it is not reported in the table

- Table 2: In the line “Referred to another hospital”, remove bold from p-value if not significant

---

## [Author Response · Author response to Decision Letter 0]

16 Feb 2022

Reviewer #1: In this prospective study, authors investigated the risk factors associated with ICU admission in COVID-19 patients and the outcomes (systemic complications, mortality, functional status at discharge) in hospitalized patients in 8 Latin American countries. Overall, the manuscript is sound and clearly written, and data are clearly presented in tables; however, authors could improve the manuscript according to the following suggestions.

Author’s response: We thank the reviewer for the detailed revision and positive feedback. We will answer your comments on the following pages. 

Comment 1- Add recent references in the first paragraph to update data about the mortality rate and main characteristics associated to severe COVID-19

Author’s response: Dear reviewer, we appreciate your advice. Following your recommendation, we have updated the data about the burden of COVID-19 and mortality rate worldwide reported and added the following references:

WHO Coronavirus (COVID-19) Dashboard [Internet]. Covid19.who.int. 2022 [cited 8 February 2022]. Available from: https://covid19.who.int/

Cifuentes-Faura J. COVID-19 Mortality Rate and Its Incidence in Latin America: Dependence on Demographic and Economic Variables. International Journal of Environmental Research and Public Health [Internet]. 2021 [cited 8 February 2022];18(13):6900. Available from: https://pubmed.ncbi.nlm.nih.gov/34199070/

Koupaei M, Naimi A, Moafi N, Mohammadi P, Tabatabaei F, Ghazizadeh S et al. Clinical Characteristics, Diagnosis, Treatment, and Mortality Rate of TB/COVID-19 Coinfectetd Patients: A Systematic Review. Frontiers in Medicine. 2021;8.

Comment 2- Lines 120-122: add reference

Author’s response: Dear reviewer, we appreciate your advice; we have added the following references to that section:

What do we mean by self-care? [Internet]. World Health Organization. 2022 [cited 8 February 2022]. Available from: https://www.who.int/reproductivehealth/self-care-interventions/definitions/en/

Self care interventions for sexual and reproductive health and rights | The BMJ [Internet]. Bmj.com. 2022 [cited 8 February 2022]. Available from: https://www.bmj.com/selfcare-srhr

Comment 3- Lines 132-133: why did you consider only treatment initiated in the ICU during the first 24 hours of hospital admission in your analysis?

Author’s response: We thank the reviewer for asking for this necessary clarification. We only consider the treatment initiated during the first 24 hours of admission because we aim to identify the characteristics of the COVID-19 patients and the factors associated with clinical outcomes directly related to the acute disease. Patients admitted to the ICU due to COVID-19 frequently develop systemic complications and require a diverse array of treatments that might impact their outcomes but are not directly related to the acute infection. Moreover, severely ill patients generally express their clinical and paraclinical characteristics during the first 24 hours of ICU admission, and these are the factors that might be treated early. Also, identifying the factors associated with worse clinical outcomes based on the characteristics gathered during the first 24 hours of admission might help clinicians identify patients who would benefit the most from being admitted to the ICU.

Comment 4- Lines 200-202: specify how many patients you included in the LOS evaluation, according to the sentence in the Methods section (lines 134-135). Data should also be reported in a table.

Author’s response: We appreciate your comment. Following the reviewer’s comment, we have included these data in the text and table 2. It now reads as follows:

“Regarding hospital length of stay (LOS), we only include 2823 patients because a total of 185 patients has missing data of discharge date; the overall median (IQR) observed in the cohort was 10 (4-19); when stratified by ICU admission, we found that ICU admitted patients had significantly longer hospital LOS (15 [9-26] vs. 3 [0-7], p<0.001).”

Comment 5- Line 231: add references

Author’s response: We have added the following reference to the sentence pointed out by the reviewer: 

Oliveira E, Parikh A, Lopez-Ruiz A, Carrilo M, Goldberg J, Cearras M et al. ICU outcomes and survival in patients with severe COVID-19 in the largest health care system in central Florida. PLOS ONE [Internet]. 2021 [cited 8 February 2022];16(3):e0249038. Available from: https://pubmed.ncbi.nlm.nih.gov/33765049/

Comment 6- Line 268: add references

Author’s response: We have modified the text to make it clear that we were talking about the same manuscript mentioned before. It now reads: “Moreover, a meta-analysis including 37 articles revealed that the pool prevalence of ICU mortality in patients with COVID-19 was 32%. This meta-analysis did a subgroup analysis by the country where the highest mortality rates were reported in China (42%), followed by the USA (36%) [37]. Our cohort found an ICU mortality rate of 35%, similar to the overall mortality presented on the metanalysis [37], though relatively lower when compared to the United States and the Italian cohorts.”

The 37 references correspond to Abate SM, Ahmed Ali S, Mantfardo B, Basu B: Rate of Intensive Care Unit admission and outcomes among patients with coronavirus: A systematic review and Meta-analysis. PLoS One 2020, 15(7):e0235653

Comment 7

- Table 1: Add CRP in the section of laboratories test on hospital admission

- Table 1: remove LOS from the acronym list because it is not reported in the table

- Table 2: In the line “Referred to another hospital,” remove bold from p-value if not significant

Author’s response: We thank the reviewer for this comment. We have edited the tables and text accordingly.

---

## [Decision Letter · Decision Letter 1]

4 Mar 2022

Clinical Characteristics, Systemic Complications, and In-Hospital Outcomes for Patients with COVID-19 in Latin America. LIVEN-Covid-19 Study: A Prospective, Multicenter, Multinational, Cohort Study.

PONE-D-21-34973R1

Dear Dr. Reyes,

We’re pleased to inform you that your manuscript has been judged scientifically suitable for publication and will be formally accepted for publication once it meets all outstanding technical requirements.

Kind regards,

Prof. Raffaele Serra, M.D., Ph.D

Academic Editor

PLOS ONE

Additional Editor Comments (optional):

amended manuscript is acceptable

Reviewers' comments:

Reviewer's Responses to Questions

**Comments to the Author**

1. If the authors have adequately addressed your comments raised in a previous round of review and you feel that this manuscript is now acceptable for publication, you may indicate that here to bypass the “Comments to the Author” section, enter your conflict of interest statement in the “Confidential to Editor” section, and submit your "Accept" recommendation.

Reviewer #1: All comments have been addressed

2. Is the manuscript technically sound, and do the data support the conclusions?

Reviewer #1: Yes

3. Has the statistical analysis been performed appropriately and rigorously? 

Reviewer #1: Yes

4. Have the authors made all data underlying the findings in their manuscript fully available?

Reviewer #1: Yes

5. Is the manuscript presented in an intelligible fashion and written in standard English?

Reviewer #1: Yes

6. Review Comments to the Author

Reviewer #1: In this revised version, authors improved the quality of the manuscript according to my suggestions. The bibliography is now sufficiently updated.

7. PLOS authors have the option to publish the peer review history of their article (what does this mean?). If published, this will include your full peer review and any attached files.

Reviewer #1: No

---

## [Editor Report · Acceptance letter]

21 Mar 2022

PONE-D-21-34973R1 

Clinical Characteristics, Systemic Complications, and In-Hospital Outcomes for Patients with COVID-19 in Latin America. LIVEN-Covid-19 Study: A Prospective, Multicenter, Multinational, Cohort Study. 

Dear Dr. Reyes:

I'm pleased to inform you that your manuscript has been deemed suitable for publication in PLOS ONE. Congratulations! Your manuscript is now with our production department. 

Kind regards, 

on behalf of

Prof. Raffaele Serra 

Academic Editor

PLOS ONE